# A Meroterpenoid from Tibetan Medicine Induces Lung Cancer Cells Apoptosis through ROS-Mediated Inactivation of the AKT Pathway

**DOI:** 10.3390/molecules28041939

**Published:** 2023-02-17

**Authors:** Yi Huang, Yun Huang, Ge Zhu, Bingzhi Zhang, Yujia Zhu, Bin Chen, Xiaoxia Gao, Jie Yuan

**Affiliations:** 1Key Laboratory of Tropical Disease Control (Sun Yat-sen University), Ministry of Education, Guangzhou 510080, China; 2School of Pharmacy, Guangdong Pharmaceutical University, Guangzhou 510006, China; 3School of Basic Medical Sciences, Southern Medical University, Guangzhou 510515, China; 4Department of Biochemistry, Zhongshan School of Medicine, Sun Yat-sen University, Guangzhou 510080, China; 5School of Public Health, Sun Yat-sen University, Guangzhou 510080, China; 6Southern Laboratory of Ocean Science and Engineering, Zhuhai 519000, China

**Keywords:** D1399, meroterpenoid, Tibetan medicine, apoptosis, ROS, AKT

## Abstract

As a traditional Tibetan medicine in China, *Meconopsis grandis* Prain has been used to treat a variety of illnesses by local people for thousands of years. However, the active ingredients contained in *Meconopsis grandis* Prain and its pharmacodynamic mechanisms have scarcely been reported. We isolated a meroterpenoid named D1399 from *Meconopsis grandis* Prain endophytic fungi with strong antitumor activity. The structure analysis showed that D1399 is an alkaloid containing a 13-membered macrocyclic structure. The IC_50_ of D1399 for human lung cancer cells’ viability ranged from 0.88 to 2.45 μM. Furthermore, we utilized TUNEL assay and western blotting to investigate the antitumor effectiveness of D1399. The results have shown that D1399 induced the apoptosis of lung cancer cells on the extrinsic and intrinsic pathways by boosting ROS generation and repressing AKT activity. In the mouse xenograft model, the average tumor weight with 30 mg·kg^−1^ D1399 treatment exhibited 73.19% inhibition compared with the untreated control, without affecting body weight loss. Above all, for the first time, our study provides a possible mechanism for the antitumor activity of D1399 in vitro and in vivo as a natural product from Tibetan medicine with *Meconopsis grandis* Prain, which may be a potentially promising antitumor drug candidate.

## 1. Introduction

In the history of more than 2000 years, traditional Tibetan medicine (TTM) has played an important role in the prevention and treatment of various diseases, including cancer [1]. *Meconopsis grandis* Prain (family *Papaveraceae*) is a unique and endangered herb used as an important form of traditional Tibetan medicine in China [2]. For thousands of years, this plant has always had very good treatment effects for a variety of diseases, including inflammation, pain, lung heat, cough, liver heat, and fever of animals and humans. Recently, some of the secondary metabolites of endophytic fungus from *Meconopsis grandis* Prain were extracted [3], but few studies have investigated the role of the secondary metabolites on antitumor activities. In this paper, D1399, a meroterpenoid containing rare 13-membered macrocycles, was isolated from the fungal strain DH2 (*Neonectria* sp.), which was obtained and identified from the roots of *Meconopsis grandis* Prain collected from Cona County, Tibet Autonomous Region, P. R. China. Study on the drug activity and molecular mechanisms of its secondary metabolisms is beneficial to developing and protecting the endangered Tibetan medicine rationally.

Meroterpenoids are a kind of natural product of mixed biosynthetic origin which are partially derived from terpenoids [4], which are often isolated from fungi, sponges, seaweed, and marine organisms. Most of them have a wide range of biological activities. Recently, antitumor activity of the meroterpenoid derivatives, including Pyrrocidine A, Pyrrocidine B [5,6,7], Pyrrocidine C [8], Hirsutellone A, Hirsutellone B, and Hirsutellone C [9], was reported. Now, meroterpenoids’ rich medicinal activities have gradually attracted the attention of researchers.

Lung cancer is the most common cause of cancer-related deaths worldwide [10]. There were more than 2.21 million new cases and 1.80 million deaths due to lung cancer in 2020 [11]. In China, lung cancer diagnosis accounts for one-third of all cancer diagnoses and has already been a financial burden on the country [12]. Treatment options for lung cancer include surgery, radiation therapy, chemotherapy, and targeted therapy [13]. Despite the improvements in therapy made during the past years, the prognosis for patients with lung cancer is still unsatisfactory. The responses to current standard therapies are poor except for the most localized cancers. Thus, to effectively treat lung cancer, the identification of new molecular targets and the development of novel targeted therapeutic agents are urgently needed.

The AKT pathway plays a crucial role in the tumorigenesis and development of lung cancer [14]. The activation of AKT by phosphorylation is associated with poor differentiation [15,16], larger tumor size [17], and an advanced stage [18,19]. Targeted agents against the AKT pathway are presently being investigated in early clinical trials for the sake of lung cancer treatment with promising results [20,21]. Both preclinical animal studies and clinical trials in humans have validated AKT as an attractive target for cancer drug discovery [22]. It has been widely proposed that reactive oxygen species (ROS) release can regulate cancer progress via repressing PI3K/AKT/mTOR pathways, such as the cell cycle, apoptosis, and autophagy [23,24]. While, under physiological conditions, the level of intracellular ROS keeps appropriate balance in redox modulation and cell growth, excessive ROS production clearly damages DNA and triggers apoptosis [25]. Interestingly, cancer cells are more sensitive to rapid increases in ROS levels than normal cells [26]. Therefore, a variety of agents that stimulate the ROS generation have been reported recently, such as Erlotinib, vemurafenib, etc., as effective strategies for treating human cancer [27,28].

Here our primary purpose was to ascertain the antitumor function of novel meroterpenoids in human cancer, and firstly to investigate the possible associated molecular mechanisms of D1399 in lung cancer cell lines A549 and H460. Our results strongly suggest that meroterpenoids containing a 13-membered macrocyclic alkaloid structure are a kind of cancer inhibitor. Our data go deeply into the matter that D1399 suppressed lung cancer cell growth as a potent apoptotic inducer by ROS-mediated negative regulation of the AKT pathway in vitro and in vivo.

## 2. Results

### 2.1. Chemistry of D1399

D1399 was obtained as a white powder. The molecular formula of D1399 was determined to be C_32_H_29_NO_4_ by HREI-MS (*m*/*z* 501.2874). The ^1^H and ^13^C NMR spectral data of D1399 are listed in Appendix A.

Based on the related literature [7,29] and NMR data of D1399, the structure of D1399 is very similar to pyrrocidine A. The molecular formula indicates one additional carbon atom and two additional hydrogen atoms to that of pyrrocidine A. The analysis of its ^1^H and ^13^C NMR spectral data led to the conclusion that this compound differed from pyrrocidine A only in the chemical shift of C-19. In the HMBC spectrum, a correlation from H-27 at δ 2.28 to C-19 at δ 92.9 implied that C-19 and C-27 were connected through an ether linkage. The DEPT spectrum showed that C-27 is an sp3 carbon signal, indicative of a methoxy replacing the hydroxyl of pyrrocidine A. Taken together, according to the NOE data (Appendix A), the relative configuration structure of D1399 was the same as pyrrocidine A except for a methoxy bonded to C-19, and D1399 was named as 19-methoxy-pyrrocidine A (Figure 1).

### 2.2. D1399 Inhibits the Growth of Human Cancer Cells

To evaluate the effect of D1399 on human lung cancer cells’ viability, several different genotype lung cancer cell lines (A549, PC9, Calu1, H1299, and H460) were treated with D1399 for 48 h. As shown in Figure 2, D1399 exhibited a significant growth inhibition on human lung cancer cells with the IC_50_ ranging from 0.88 to 2.45 μM (the IC_50_ was 1.18 μM for A549, 1.64 μM for PC9, 0.88 μM for Calu1, 1.25 μM for H1299, and 2.45 μM for H460, respectively). Moreover, we also tested the effect of D1399 on more human cancer cells’ viability. As expected, D1399 could inhibit the proliferation of different cancer cell lines, including breast cancer (MDA-MB-231, MCF7), melanoma (MDA-MB-435), liver cancer (HepG2), and colon cancer (HCT-116) (Appendix A). 

### 2.3. D1399 Induces Caspase-Dependent Apoptosis in Lung Cancer Cells

Cell viability is a dynamic process that reflects a balance between cell growth and cell death. To find out whether the inhibition of cell proliferation by D1399 is associated with the induction of apoptotic cell death, we characterized changes in the chromatin morphology of cells using DAPI staining. To confirm the induction of the apoptotic cell death by D1399, we performed TUNEL staining assays, and then the labelled cells were investigated using fluorescence microscopy. Cells were labelled by the TUNEL assay, and both H460 and A549 cells showed an increase in fluorescence intensity in the presence of D1399 (Figure 3A). The number of apoptotic cells (DAPI-positive TUNEL-stained cells) presented a significant increase by the treatment of increasing concentrations of D1399 in H460 and A549 cells (Figure 3B). The results showed that D1399 could increase apoptotic cells in a dose-dependent manner. Additionally, the death of human lung cancer cells treated with D1399 was a consequence of cell apoptosis, caused by the compound as an apoptosis inducer.

As the activation of caspases is a significant mechanism for apoptotic cell death [30], the further characteristics of the apoptotic pathways involved in D1399 were tested by western blotting, especially about the crucial caspases. As represented in Figure 3C,D, D1399 treatment dose-dependently brought about activating cleavage of caspase-8 and caspase-9, concurrently displaying a significant increase of cleaved bands, accompanied by a reduction in the level of the procaspase-8 and procaspase-9. These results indicated that both caspase-8- and caspase-9-dependent apoptotic pathways were activated by D1399. Caspase-3, another major effector caspase, was also examined. Cleaved, activated caspase-3 was detected in H460 cells and A549 cells after D1399 treatment. Cleavage of PARP was also clearly demonstrated with D1399 treatment both in H460 cells and A549 cells. These data indicated that the apoptosis induced by D1399 was relevant to both intrinsic and extrinsic pathways in lung cancer cells.

### 2.4. D1399 Increases the ROS Generation and Represses Activated AKT in Lung Cancer Cells

ROS plays an important role in apoptosis induction and mediates caspase activation as an apoptosis trigger directly or indirectly [31]. There are also reports that intracellular ROS generation is crucial for chemotherapeutic agent-induced apoptosis in various cancer cells [32]. Thus, to analyze whether D1399-induced apoptosis is mediated through ROS excessive generation, the H460 and A549 were treated with D1399 in a dose-dependent concentration, and the change of ROS levels by the use of the redox-sensitive fluorescence probe DCFH-DA was observed (Figure 4A,B). Using H_2_DCFDA-based detection and flow cytometry, ROS accumulation was observed after treating cells with D1399 for 4 h. The results showed that D1399 induced significantly higher levels of intracellular ROS in a dose-dependent manner (Figure 4C). However, cells co-treated with *N*-acetyl cysteine (NAC), a potent ROS scavenger, showed significantly reduced ROS levels compared to those of D1399 alone-treated cells (Figure 4D), which confirmed that D1399 resulted in intracellular ROS accumulation.

In the past studies, it has been confirmed that the release of ROS has an important impact on the AKT signaling pathway, which is involved in the apoptosis of tumor cells [33]. AKT has a consequential role in promoting cell survival downstream of oncogenes, cell stress molecules, and growth factors [34]. From a therapeutic point of view, AKT is an important target in human lung cancer treatment. To explore the role of the AKT pathway in D1399-induced apoptosis of H460 cells and A549 cells, we tested the phosphorylation level of AKT under the treatment of D1399. The western blotting results showed that the levels of AKT phosphorylation at Ser473 and Thr308 markedly decreased in a dose-dependent manner, but total AKT levels were unaffected following (Figure 5A,B). We further investigated the role of the AKT signaling pathway on ROS generation-mediated apoptosis by D1399. Figure 5C shows that when the production of ROS was artificially blocked, the reduced phosphorylation levels of AKT by D1399 were maintained at the control level. In addition, we also had some reasonable observational support that D1399 affected the expression status of proteins downstream of AKT. AKT has been found to directly phosphorylate S196 on human procaspase-9, and this phosphorylation correlates with a decrease in the protease activity of caspase-9 in vitro [34]. Our results demonstrated that the expression levels of caspase-9 were decreased and cleaved caspase-9 were increased in response to D1399 in H460 cells and A549 cells (Figure 3C,D). To make certain the influence of D1399 on the AKT pathway, A549-myr-AKT cells (AKT is highly expressed) and A549-WT cells (AKT is normally expressed) were treated by D1399. Apparently, IC_50_ for A549-myr-AKT cells was higher than IC_50_ for A549-WT cells (Figure 5D). It showed that the AKT pathway was significant for D1399-induced apoptosis in A549. Considering the above evidence, we could assert that a deviant AKT signaling pathway activates caspase-9 and contributes to D1399-induced cell apoptosis.

### 2.5. D1399 Inhibits the Growth of Lung Tumor Cell Xenografts

To confirm the antineoplastic activity of D1399 in vitro, BALB/c athymic nude mice were used to establish xenografts of lung tumor cells by being subcutaneously inoculated with H460 cells. As the tumor volumes and the general status of the mice were assessed after receiving the vaccination on day five, three groups of mice were found to have developed S.C. tumors. Following i.p. injection with D1399 (15 and 30 mg·kg^−1^) every three days, a remarkable inhibition of the growth of the xenografted tumors with 30 mg·kg^−1^ D1399 was observed. D1399 effectively suppressed the volumes (Figure 6A,B) of H460 tumors implanted in BALB/c athymic nude mice. At the end of the experiment, the average tumor weight with 30 mg·kg^−1^ D1399 treatment exhibited 73.19% inhibition compared with the untreated control (Figure 6C) without affecting body weight loss (Figure 6D).

## 3. Discussion

*Meconopsis grandis* Prain is a kind of traditional Tibetan medicine with rich metabolic products. In general, it is a perennial herb from 40 cm to 120 cm tall and grows in alpine scrub and alpine meadows with shady and half-shady slopes at altitudes of 3000–5500 m. Most of them are distributed in the south-central part of Tibet [2]. Due to its rarity in the nature and difficulty in culture, the plant has become an endangered species. This is one of the major bottlenecks for the activity research of *Meconopsis grandis* Prain as a medicinal plant. Therefore, it is an effective way to reduce the destruction of wild *Meconopsis grandis* Prain while investigating the pharmaceutically active metabolites extracted from the fermented product of endophytic fungi of *Meconopsis grandis* Prain for studies on its medicinal properties. Recently, the secondary metabolites of *Meconopsis grandis Prain* were reported, and the results showed that three polyphenols were extracted from the fermentation compounds of the endophytic fungi isolated from *Meconopsis grandis* Prain, including a new natural product with a certain complexation ability for iron [3]. Up until now, however, little literature is about the systematic studies on antitumor activities of the secondary metabolites from *Meconopsis grandis* Prain or its endophytic fungi. In this paper, for the first time, D1399 was purified from Tibetan *Meconopsis grandis* Prain endophytic fungus *Neonectria* sp., which contains a unique 13-membered macrocyclic alkaloid structure. In this context, we revealed that D1399 had in vivo and in vitro antitumor activity. The possible molecular mechanism underlying the role of D1399 in the inhibition of the growth of lung cancer cells is that D1399 increased ROS generation and reduced the phosphorylation of AKT, thus inducing cell apoptosis via both the extrinsic and intrinsic apoptotic signal pathways (Figure 6E). In addition, D1399 showed effectiveness in inhibiting xenografted tumor growth with a treatment of 30 mg·kg^−1^ D1399 in vivo in this study. The pharmacological and toxicological profiles revealed that D1399 has a major probability to become an attractive antitumor drug candidate.

As our data provided, D1399 widely inhibited various human cell lines, including five human lung cancer cells, A549, PC9, Calu1, H1299, and H460, with IC_50_ < 2.45 μM. It is well known that apoptotic susceptibility of the tumor cells is a key determinant of the efficacy of chemotherapy. The induction of apoptosis is an efficient mechanism of antitumor agents, especially natural products [35]. More importantly, some TTMs and their active ingredients have been reported to possess anticancer activity by targeting some apoptosis pathways in cancer [36]. Hence, the effect of apoptosis induced by D1399 was investigated firstly. In current study, the results of fluorescence microscopic examination revealed that D1399 induced the apoptosis in H460 and A549 cells, which were presented as the TUNEL-positive cells increasing with the increasement of the dose. In previous studies, A549 was found to acquire drug resistance due to a mutation in the *KRAS* gene [37], and H460 respectively has two mutations, which include the *KRAS* gene and the *PIK3CA* gene. Additionally, various studies have indicated *KRAS* gene and *PIK3CA* gene mutations will give rise to larger and more advanced tumors [38]. Thus, whether D1399-induced apoptosis is associated with the mutation of the *KRAS* gene and *PIK3CA* gene in H460 and A549 cells still requires further study.

A compelling study suggests that ROS has an important relation with apoptosis, depending on the activation of the extrinsic and intrinsic pathways [39]. On the one hand, ROS can downregulate the c-FLIP half-life by inducing its ubiquitin-proteasomal degradation, therefore enhancing this extrinsic pathway [32]. On the other hand, many ROS-related anticancer drugs could increase the cytoplasmic release of pro-apoptotic factors such as pro-caspase-9 to build the apoptosome and activate the effector caspases [40]. Therefore, ROS is an important part of studying anticancer drugs. In this paper, it appears to be a dose-related increase in the oxidative stress in H460 and A549 apoptotic cells with the treatment of D1399. Moreover, caspase-9 activation as represented by the level of the 35 and 37 kDa cleaved fragments could be induced dose-dependently by D1399. These results suggest that the apoptosis of lung cancer cells mediated by caspase-9 may be caused by upregulation of ROS by D1399.

Previous studies have shown that the AKT signaling pathway has an influential role, not only in tumorigenesis and development, but also in tumor therapy strategies [41]. Therefore, AKT has been identified as an attractive therapeutic target for cancer therapy by the inhibition of AKT phosphorylation alone or in combination with standard cancer chemotherapeutics [22]. In lung cancer therapy, the treatment plan focusing on the AKT pathway is considered to be full of great challenges, as well as numerous opportunities [42]. Our results indicated that D1399 could suppress phosphorylation of AKT at both Ser473 and Thr308, and that the non-phosphorylation level of AKT remained constant in all tests. Meanwhile, in A549-myr-AKT cells with overexpression of myristoylated, the active form of AKT, as predicted, the IC_50_ of D1399 is significantly higher than that in parental A549 cells. Our data suggests that AKT is involved in the D1399-induced apoptosis of lung cancer cells. Furthermore, AKT activity is also inhibited by the release of ROS, which affects cell cycle, autophagy, and apoptosis [24]. Therefore, we hypothesized that the apoptosis of lung cancer cells induced by D1399 was caused by an ROS-dependent AKT signaling pathway.

In our study, we found that D1399 has a special 13-membered macrocyclic alkaloid structure, which may be a promising antitumor lead compound in other previous literature. Some of the compounds that contain similar 13-membered macrocyclic alkaloid structures, derived from other natural resources, for example, Pyrrocidine A, Pyrrocidine B [5,6], Pyrrocidine C [8], Hirsutellone A, Hirsutellone B, and Hirsutellone C [9], exhibit various biological activities, including antitumor effects. Moreover, hundreds of fungal meroterpenoids with a wide range of biological activities have been reported [43,44]. In addition to antitumor activities [45], these compounds also have antibacterial activity [46], insecticidal activity [47], anti-oxidative activity [46], and anti-virus activity [48]. For example, Pyripyropenes have been used as clinical cholesterol acyltransferase inhibitors [44]. These research data and clinical cases indicate the potential for the further development of meroterpenoids for disease treatment and encourage further study about D1399, not only in the antitumor field, but in other disease therapeutic areas, too. 

## 4. Materials and Methods

### 4.1. Materials and Physical Measurements

The NMR spectra were recorded on Bruker AVANCE 400 (Bruker Co. Ltd., Zurich, Switzerland). TLC was carried out on precoated silica gel GF-254 plates (Qingdao Haiyang Chemical Co., Ltd., Qingdao, China) and chromatography was performed over silica gel (200–300 mesh and 300–400 mesh; Qingdao Haiyang Chemical Co., Ltd., Qingdao, China) and Sephadex LH-20 (GE healthcare, Buckinghamshire, UK). HREIMS data were measured on a MAT95XP high-resolution mass spectrometer (Thermo Fisher Scientific Inc., Lafayette, CO, USA), and EIMS on a DSQ EI-mass spectrometer (Thermo Fisher Scientific Inc., Lafayette, CO, USA). Optical rotations were determined with an Abbemat 300 polarimeter (Anton Paar, Ashland, VA, USA) at 20 °C. UV spectra were obtained with UV BlueStar A (GenTech Scientific LLC, Arcade, NY, USA).

### 4.2. Source and Fermentation of Fungus

The strain DH2 was isolated from the roots of *Meconopsis grandis* Prain sample collected from Cona County, Tibet Autonomous Region, P. R. China and was identified as *Neonectria* sp. by ITS-rRNA gene sequence analysis. Strain DH2 was inoculated in 1000 mL baffled flasks containing 500 mL PDB seed medium (3% sea salt) and was cultured for 3 days on a rotary shaker (100 rmp). Then, 10 mL of the resulting culture was respectively transferred into 100 bottles of 1000 mL Erlenmeyer flasks containing rice medium (60 mL rice, sea salt 3% in 80 mL water) and fermented by static culturing for 28 days.

### 4.3. Extraction and Isolation

The fermentation medium was immersed in a mixed solution of methanol, ethyl acetate, and chloroform (*v*/*v*/*v* = 1:1:1, 4.0 L × 3) for 3 days and filtered through a filter to obtain the solution, which was concentrated under reduced pressure and extracted with ethyl acetate (3.0 L × 4). The ethyl acetate crude extract (50 g) was subjected to column chromatography over silica gel eluting with a gradient of petroleum ether–ethyl acetate from 100:0 to 0:100 (*v*/*v*), and four fractions were collected. Fraction A (965 mg) was further subjected to Sephadex LH-20 column chromatography eluted with CH_2_Cl_2_:MeOH (1:1) to give fraction A-1. Fraction A-1 (545 mg) was purified by a silica gel column with petroleum ether–ethyl acetate (3:2) to give fraction A-1-4 (340 mg), and then it was applied to Sephadex LH-20 column chromatography eluting with CH_2_Cl_2_:MeOH (1:1) to obtain D1399 (112.5 mg). D1399 was a 1 mM stock solution dissolved in DMSO (Dimethylsulphoxide), which we need to use every time, diluted on the basis of the experimental requirements.

### 4.4. Cell Culture

Human lung cancer cell lines H460, A549, Calu1, H1299, and PC9, breast cancer cell lines MDA-MB-231 and MCF-7, melanoma cell line MDA-MB-435, liver cancer cell line HepG2, and colon cancer cell line HCT-116 were purchased from cell banks of Shanghai Institutes of Biological Sciences (Shanghai, China) or from Fu Erbo Biotechnology Co., Ltd. (Guangzhou, China). A549-myr-AKT is A549 cells transfected with active AKT (myr-AKT), which was a kind gift from professor Junchao Cai (Sun Yat-sen University, Guangzhou, China) [49]. These cell lines were maintained in DMEM medium (Invitrogen, Carlsbad, CA, USA) and supplemented with 10% FBS (Hyclone, Logan, UT, USA), 1% penicillin/streptomycin (Invitrogen, Carlsbad, CA, USA), and 2 mM l-glutamine. Cells were passaged with 0.05% Trypsin (Life Technologies, Carlsbad, CA, USA) every 2~3 days and were cultured in the cell culture CO_2_ incubator at 37 °C.

### 4.5. Cell Growth Assay

A total of 1 × 10^4^ cells were seeded per well in 96-well flat-bottom plates cultured overnight, and were respectively treated with various concentrations of D1399 for 48 h. Then, we added 28 μL 5 mg·mL^−1^ concentration of 3-(4,5-dimethylthiazol-2-yl)-2,5-diphenyl tetrazolium bromide (MTT) (Genview, Houston, TX, USA) as a work solution into each well and returned the 96-well plate to a humidified CO_2_ tank for 4 h. Finally, 200 μL DMSO (Sangon Biotech, Shanghai, China) was added to every well and allowed to dissolve the MTT-formazan crystals formed. The optical density (OD) was then measured at 490 nm with a reference wavelength of 630 nm using a microplate spectrophotometer.

### 4.6. Apoptosis Analysis by TUNEL

TUNEL assay was performed using the Dead End™ fluorometric TUNEL assay kit. The TUNEL assay for in situ detection of apoptosis was performed by using the Dead End™ Fluorometric TUNEL System assay kit (Promega, Madison, WI, USA), according to the manufacturer’s instructions. H460 and A549 cells were treated with D1399 at concentrations of 0.5 × IC_50_, 1.0 × IC_50_, and 1.5 × IC_50_ for 24 h. Following, cells were fixed in 4% paraformaldehyde solution at 4 °C for 25 min. Fixed cells were washed three times with 1 × PBS for 5 min and then permeabilized in 0.1% Triton X-100 for 15 min. After rinsing the slides with 1 × PBS, cells equilibrated in an equilibration buffer for 10 min and treated terminal deoxynucleotidyl transferase (TdT) was stored away from light for 1 h at 37 °C to label with fluorescein-12-dUTP. After a wash with 2 × SSC for 15 min at room temperature, the cells were treated with 1 mg·mL^−1^ DAPI solution for 15 min. The slides were observed by fluorescence microscopy with an inverted microscope (Zeiss Axiovert100M, Carl Zeiss, Germany). A total of ten randomly chosen microscopic fields, including green fluorescence of apoptotic cells, were captured and calculated. Experiments were performed in triplicate.

### 4.7. Western Blotting Analysis

H460 and A549 cells were treated with D1399 at various concentrations of 0.5 × IC_50_, 1.0 × IC_50_, and 1.5 × IC_50_, or pre-treated with 2 mM *N*-acetyl-l-cysteine (NAC) (Sigma-Aldrich Pty Ltd., Darmstadt, Germany) for 1 h before D1399 treatment. After about 6 h, cells, including dead cells, were collected and washed with PBS and lysed in 1× sampling buffer. Proteins recovered by mild sonication and protein concentrations were then determined using the Bradford assay. An aliquot of the denatured supernatant containing 40 mg of protein was resolved on SDS-PAGE and transferred to nitrocellulose membranes (Bio-Rad, Hercules, CA, USA). After blocking with a blocking buffer (Tris-buffered saline, i.e., TBS, containing 5% non-fat milk) for 1 h at room temperature, the membrane was probed with primary antibodies for mouse anti-human caspase-8 (BD Biosciences, San Jose, CA, USA), rabbit anti-human caspase-3, rabbit anti-human poly (ADP-ribose) polymerase (PARP), rabbit anti-human phospho-AKT (Ser473), rabbit anti-human caspase-9, rabbit anti-human AKT, rabbit anti-human PDK-1, rabbit anti-human GAPDH (Cell Signaling Technology, Beverly, MA, USA), and rabbit anti-human phospho-AKT (Thr308) (Abcam, Cambridge, MA, USA), and then incubated overnight at 4 °C. Then, they received further incubation with the appropriate secondary antibodies for 2 h at room temperature. Protein expression was examined using an ECL system (Amersham Pharmacia Biotech, Inc., Piscataway, NJ, USA).

### 4.8. Chemicals and Fluorescent Probes for Studying ROS Generation

3 × 10^4^ A549 and H460 cells were plated on 24 mm dishes, allowed to attach overnight, and then treated with D1399 at concentrations of 0.5 × IC_50_, 1.0 × IC_50_, and 1.5 × IC_50_ for 4 h or pre-treated with 2 mM NAC for 1 h before D1399 treatment. Cells were incubated with 10 μM DCFH-DA (Sigma-Aldrich Pty Ltd., Darmstadt, Germany) without FBS at 37 °C with 5% CO_2_ for 30 min in the dark. Then, cells were centrifuged and resuspended in PBS before adding 10 μg/mL Hoechst33342 in DMEM without FBS, and incubated again at 37 °C with 5% CO_2_ for 20 min in the dark. Cells were centrifuged and washed by PBS, and this process was repeated three times. Finally, DCF fluorescence (produced in the presence of ROS) was analyzed using an epifluorescence microscope equipped with a digital camera (Zeiss, Germany). The wavelength Ex/Em = 355/465 nm (Hoechst 33342) and Ex/Em = 488/525 nm (FITC). 

### 4.9. Measurement of Intracellular ROS Level 

Intracellular ROS were measured by flow cytometry utilizing DCFH-DA. Firstly, 5 × 10^5^ A549 and H460 cells were plated on 60 mm dishes, allowed to attach overnight, and then treated with D1399 for 4 h. Cells were incubated with 10 μM DCFH-DA (Sigma-Aldrich Pty Ltd., Darmstadt, Germany) without FBS at 37 °C with 5% CO_2_ for 30 min in the dark. Finally, cells were centrifuged and resuspended in PBS before flow cytometric analysis of intracellular fluorescent DCF. The data was analyzed by Kaluza 2.1.1.

### 4.10. Xenografted Tumor Model and Antitumor Effect of D1399 In Vivo

Female BALB/c athymic nude mice (18~20 g, 4~5 weeks of age) were purchased from Hunan SJA Laboratory Animal Co. Ltd. On day zero, human tumor xenografts were established by injecting H460 cells (1 × 10^6^ cells in 0.1 mL per mouse) s.c. into the right side of the axillary of the mice. On day five, the formed tumors were measured, and 18 mice were randomly grouped into the control group and treatment group that administered i.p. doses of 30 and 15 mg·kg^−1^ body weight of D1399 every three days. 

The tumor volumes (width × width × length × p/6) were measured and body weights were monitored every three days during the treatment. Data were shown as the means ± SD. The mice were killed and measured in the experiment after the end for the tumor’s weight. All animal care and experimental procedures were approved by the Institutional Animal Care and Use Committee of Sun Yat-sen University.

### 4.11. Statistical Analysis

All of the data were representatives from at least three independent experiments. Data were analyzed with the GraphPad Prism^®^ program (GraphPad Software Inc., San Diego, CA, USA) using Student’s *t*-test or one way ANOVA methods and were expressed as the means  ±  SD. *, *p*  <  0.05, **, *p*  <  0.01, ***, and *p*  <  0.001 were indicated to be statistically significant.

## Figures and Tables

**Figure 1 molecules-28-01939-f001:**
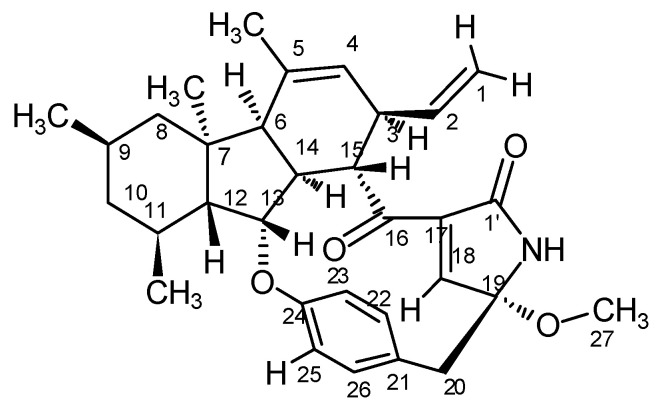
The relative configuration of D1399.

**Figure 2 molecules-28-01939-f002:**
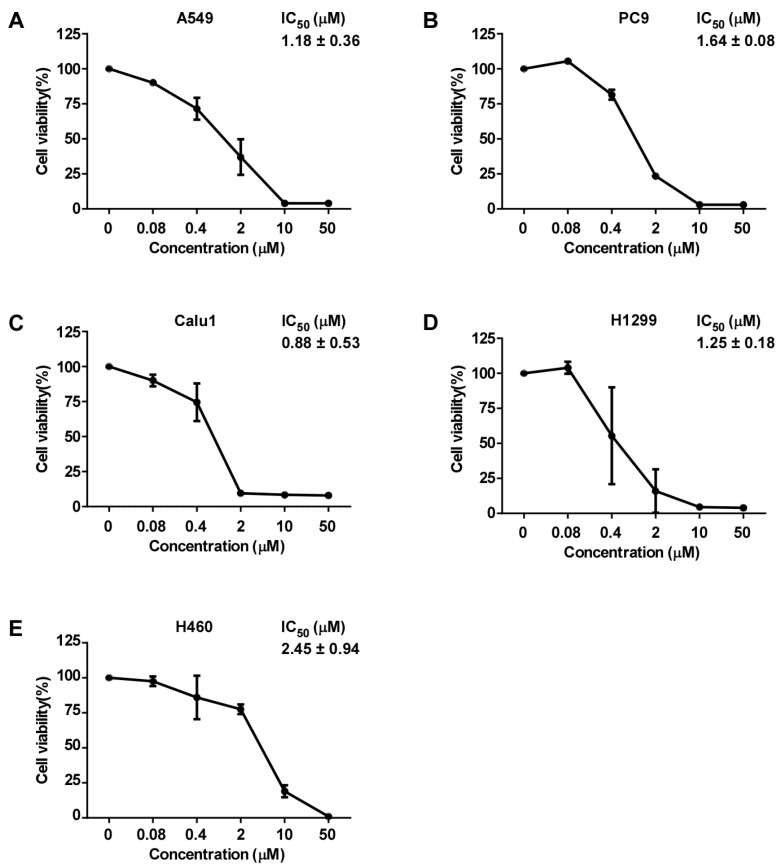
Growth inhibition of human lung cancer cells by D1399. Dose-response curves of D1399 against lung cancer cell lines, which included A549 (**A**), PC9 (**B**), Calu1 (**C**), H1299 (**D**), and H460 (**E**). IC_50_ values of D1399 on the listed lung cancer cell lines. Cells were treated with the indicated concentrations of D1399 (0 μM, 0.08 μM, 0.4 μM, 2 μM, 10 μM, 50 μM) for 48 h. Cell viabilities were examined by MTT assay. Data points are shown as means ± SD of triplicate experiments.

**Figure 3 molecules-28-01939-f003:**
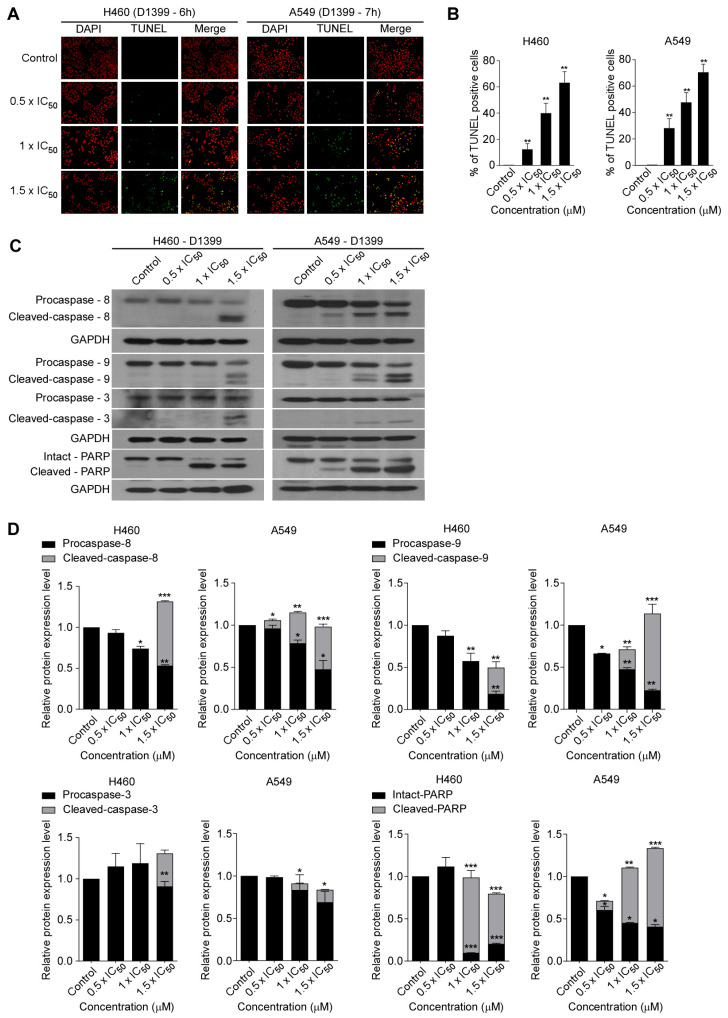
D1399-induced apoptosis in lung cancer cells. (**A**) H460 and A549 cells were respectively treated with various concentrations of D1399 for 6 h and 7 h and labelled with fluorescein-12-dUTP (green) and DAPI (red); (**B**) Apoptotic index as examined by counting and calculating the percentages of TUNEL-positive cells in 10 fields; (**C**) Influence of D1399 on the expression of apoptosis regulatory proteins. Western blotting analysis of caspases and PARP in H460 and A549 cells after D1399 treatment at the indicated concentrations and using antibodies against caspase-8, 9, and 3 and the poly adenosine diphosphate-ribose polymerase (PARP) is shown. The glyceraldehyde-3-phosphate dehydrogenase GAPDH is used as control; (**D**) Quantification shows the influence of D1399 on the expression of apoptosis regulatory proteins, including caspase-8, 9, and 3 and PARP. All protein levels were normalized to the GAPDH protein level. Data for the quantitative assessment of the proteins are expressed as means ± SD. * *p* < 0.05, ** *p* < 0.01, *** *p* < 0.001.

**Figure 4 molecules-28-01939-f004:**
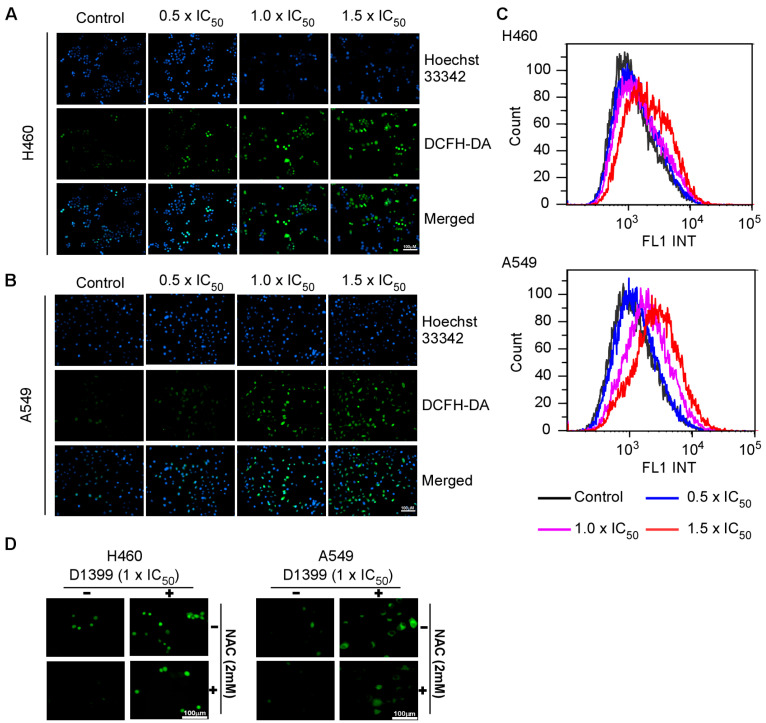
D1399-induced ROS generation in H460 and A549 cells. The cells were treated with D1399 at concentrations of 0.5 × IC_50_, 1.0 × IC_50_, and 1.5 × IC_50_ for 4 h, and the living cells and ROS were labeled with Hocchst33342 and DCFH-DA, respectively. (**A**,**B**) Fluorescence micrograph of H460 (**A**) and A549 (**B**) cells labeled by Hocchst33342 and DCFH-DA; (**C**) The ROS accumulation in H460 and A549 was observed by flow cytometry; (**D**) Fluorescence micrograph of H460 and A549 cells labeled by DCFH-DA. The cells were either treated with D1399 at concentrations of 1.0 × IC_50_ for 4 h or pre-treated with 2 mM NAC for 1 h before D1399 treatment. The images presented here are captured from one experiment, and are representative of at least three independent experiments.

**Figure 5 molecules-28-01939-f005:**
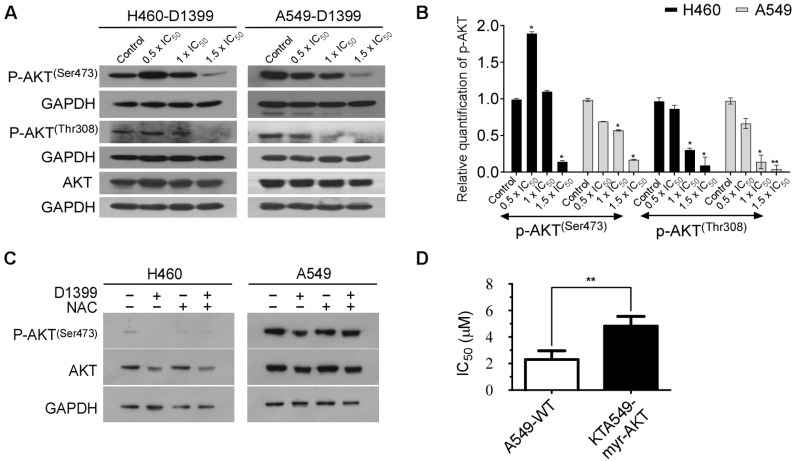
D1399 induces the ROS-dependent inactivation of the AKT signaling pathway in H460 and A549 cells. (**A**) The levels of AKT phosphorylation at Ser473 and Thr308 and total AKT were tested under the treatment of D1399 at indicated concentrations by western blotting in H460 and A549 cells. GAPDH is used as the control; (**B**) Bar graphs illustrating the relative expression levels of p-AKT proteins quantified by western blotting. All protein levels were normalized to the GAPDH protein level; (**C**) The levels of AKT phosphorylation at Ser473 and total AKT were tested by western blotting in H460 and A549 cells. The cells were either treated with D1399 for 6 h or pre-treated with 2 mM NAC for 1 h before D1399 treatment. GAPDH is used as the control; (**D**) The IC_50_ of A549-WT and A549-myr-AKT under the treatment of D1399 are compared. Results are expressed as arbitrary units. Data for the quantitative assessment of the proteins are expressed as means ± SD. *, *p* < 0.05, **, *p* < 0.01.

**Figure 6 molecules-28-01939-f006:**
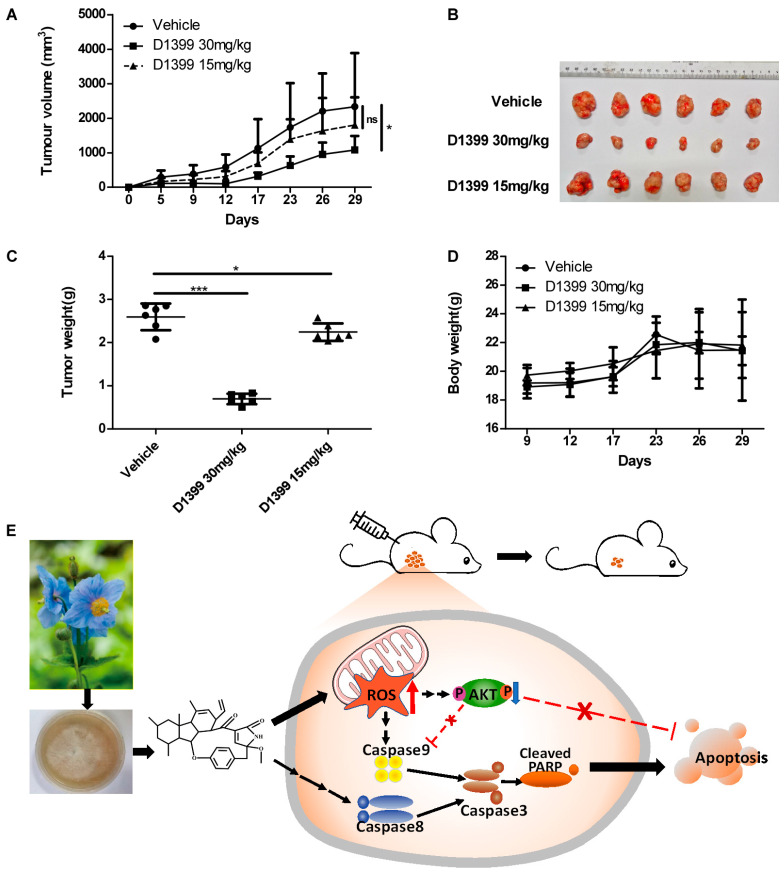
Antitumor effect of D1399 in vivo on xenografted H460 tumors in nude mice. Mice were inoculated with H460 cells (1 × 10^6^ cells per mouse) in the right side of the axillary. When tumor volume reached 80–200 mm^3^, mice received intraperitoneal injection of D1399 (15 mg·kg^−1^ and 30 mg·kg^−1^) every three days, n = 6 mice per group. The statistical comparison vs. vehicle-treated control is shown by a *t*-test. * *p* < 0.05, *** *p* < 0.001, ns, Non-Significant. (**A**) The average volume and standard deviation were recorded and plotted; (**B**) The subcutaneous tumors formed by the H460 cells were dissected and imaged; (**C**) The tumor weights at the end of the experiment; (**D**) The body weight change curves of each group of mice; (**E**) A schematic diagram of the proposed mechanism of D1399 inducing cancer cell apoptosis through the ROS-AKT pathway. D1399 increases ROS generation and reduces the phosphorylation of AKT, thus inducing cell apoptosis via both caspase-8- and caspase-9-dependent apoptotic pathways. We hypothesized that D1399 induces the apoptosis of lung cancer cells on the extrinsic and intrinsic pathways by an ROS-dependent AKT signaling pathway.

## Data Availability

Not available.

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
