# Peer review of "A Meroterpenoid from Tibetan Medicine Induces Lung Cancer Cells Apoptosis through ROS-Mediated Inactivation of the AKT Pathway"

_molecules, 2023, doi:10.3390/molecules28041939_

Round 1

Reviewer 2 Report

The article “A Meroterpenoid from Tibetan Medicine Induce Cancer Cells Apoptosis through ROS-Mediated Inactivation of the AKT Pathway” by Huang et al aims at extracting Meroterpenoid from

 Meconopsis grandis Prain endophytic fungi that has antitumor activity. They used several cancer cell lines and obtained the dose response curve and showed that this drug is effective to kill the cancer cells. They also used this drug in mouse model and showed it is effective in vivo as well. They also presented the mechanism of action of the drug.

I think this may be important article and this drug may be useful in treating different type of cancers. The manuscript has poor English and needs professional editing.

After the editing I recommend publishing this article.

Reviewer 3 Report

The author's manuscript on 'A Meroterpenoid from Tibetan Medicine Induce Cancer Cells 2 Apoptosis through ROS-Mediated Inactivation of the AKT 3 Pathway' is well written but needs more experiments to support D1399's antitumor mechanism of action. So this manuscript required a major revision before accepting it for publication. The specific comments are s follows:

1.     ROS is a general term, and authors need to be specific. For example, DFHDA can only detect hydrogen peroxide. Also, several papers are pointing that DCFHDA is not a very accurate probe to measure H2O2 and came with better probes (Aplex Read) (Murphy, M.P., Bayir, H., Belousov, V. et al. Guidelines for measuring reactive oxygen species and oxidative damage in cells and in vivo. Nat Metab 4, 651–662 (2022). https://doi.org/10.1038/s42255-022-00591-z).  Authors need to include assay-specific controls for the ROS assay. For example, treating D1399 treated cells with NAC or membrane permeable catalase and measure the DCFHDA fluorescence.

2.     In the discussion, the authors proposed ROS-mediated induction of apoptosis as a possible mechanism for the antitumor activity of D1399. To support this hypothesis, the co-treatment of ROS quenchers along with D1399, and looking at apoptotic makers strongly support their hypothesis. Similar experiments are needed to support ROS causing the hypo phosphorylation of AKT.

3.     Authors need to include a Cytochrome-C release assay to support the overall hypothesis.

4.     The authors presented a diagram of the proposed mechanism, which needs to be explained in the figure legends to better understand the overall hypothesis.

In Figure 5B, the tumor volume of the last two tumors presented in the figure (from right) looks the same, but in the graph (Figure 5A), the authors showed a significant reduction in the tumor volume. Please explain the disparity or mention it in the results.  

Round 2

Reviewer 3 Report

The Authors have addressed all my concerns and accepted for publication.